# A Cascaded 3D Segmentation Model for Renal Enhanced CT Images

Dan Li[1,2#], Zhuo Chen[2#], Haseeb Hassan[1,3], Weiguo Xie[2], Bingding Huang[*1]

[1]College of Big data and Internet, Shenzhen Technology University, Shenzhen, China.
[2]Wuerzburg Dynamics Inc, Shenzhen, China.
[3]Guangdong Key Laboratory for Biomedical Measurements and Ultrasound Imaging, School of Biomedical Engineering, Shenzhen University
Health Science Center, Shenzhen, China.
#These authors contribute equally to this work
*Corresponding author: Bingding Huang (huangbingding@sztu.edu.cn)

**Abstract.** In order to compete in the KiTS21 challenge, we propose a 3D deep learning cascaded model for the renal enhanced CT image segmentation. The proposed model comprises two stages, where stage 1 segments the kidney and stage 2 segments the tumor and cyst. The proposed deep learning network architecture is based on the residual and 3D UNet architecture. The designed network is utilized for each segmentation stage (for stage 1 and stage 2). Our intended cascaded model achieved a dice score of 0.96 for the kidney, 0.81 for the tumor, and 0.45 for the cyst on the KiTS21 validation dataset.

**Keywords:** Renal segmentation; renal tumor segmentation; renal cyst segmentation

## 1 Introduction

Every year around 400,000 people are affected by kidney tumors. Due to the wide variety in kidney and kidney tumor morphology, there is currently a great interest in tumor morphology and its surgical outcomes [1]. For instance, focusing on kidney morphology is essential to advance surgical planning. In order to accelerate such research and development, KiTS challenge has been introduced. In KiTS 2019, the initial focus was on kidney and kidney tumor segmentation [2]. However, the ongoing KiTS 21 challenge focuses on three-class segmentation tasks: kidney, kidney tumor, and cyst segmentation. Automatic semantic segmentation is a promising tool for these endeavors, but morphological variability is not easy. Therefore, there is a need for reliable segmentation techniques that can perform well and provide aid to renal surgical planning. Here we propose a cascaded 3D model that efficiently performs the renal, tumor, and cyst segmentation to solve the challenging task of morphological variability.

## 2 Methods

Our intended segmentation approach consists of a two-stage segmentation model based on 3D-Unet. For this purpose, we adopt different pre-and post-processing strategies. First, we resample the original CT scans and then crop the center parts of the CT scans. The cropped CT scans are then used as the input to the segmentation network, which is the first stage of our proposed model for renal contour prediction.

Further, a histogram equalization [3] approach is applied on segmented/predicted renal to enhance the contained tumor and cyst. Because the histogram equalization uplift the pixel brightness (contrast) and useful for the subsequent segmenting step [4]. Afterward, we concatenate the segmented renal (from stage 1) and enhanced renal (after the enhancement process) as two channels and provided that as input to stage 2 (second segmentation network). Upon providing the concatenated two channels input to the stage 2 network, the intended model predicts the tumor and cyst. Finally, the prediction results of both stages are merged into a single channel and reduced to the original CT size to evaluate the model performance. The algorithmic flow of the proposed pipeline is depicted in Fig. 1.

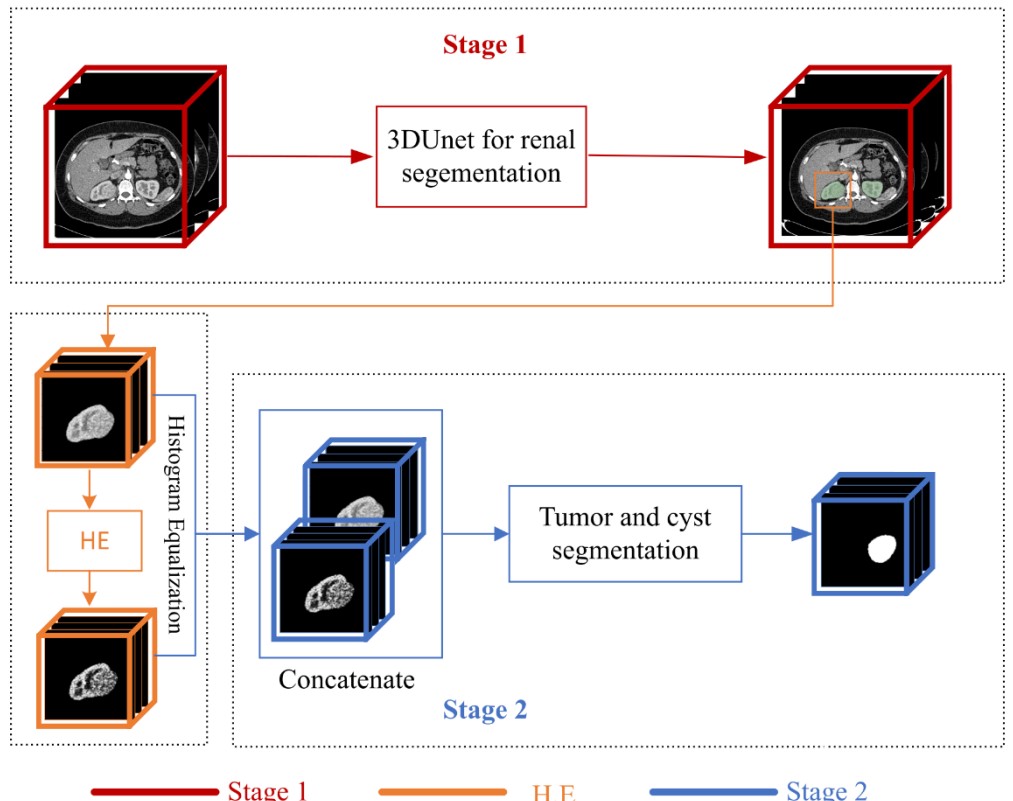

**Fig. 1.** Our proposed cascaded 3D model involves two stages. Stage 1 segments the renal, and stage 2 performs tumor and cyst segmentation. Finally, all the predicted outputs from each stage are merged into a single channel output.

## 2.1 Training and Validation Data

To train our model, we adopt the KiTS21 benchmark dataset. The benchmark dataset contains 300 contrast-enhanced CT scans, which provide three-class labels for kidney, kidney tumor, and cyst. We divide the KiTS21 dataset into training and validation sets, i.e., 270 and 30, respectively. In addition, we use voxel-wise majority voting (MAJ) for training and validation.

## 2.2 Preprocessing

As an initial preprocessing step, the CT intensities (Hounsfield units-HU) of training and validation sets were selected to a range of [-135, 215]. Doing so will change the appearance of the picture to highlight particular structures [8]. Further, we analyzed the spacing distribution of KiTS 21 CT images. After analyzing them, we found different z-axis spacing between cases, i.e., in the range [0.5mm-5mm]. To balance the z-axis spacing between each case, we resample the data and set the z-axis spacing to 3.0mm. We also use min-max normalization to normalize the HU intensities. For kidney segmentation stage 1, we resize the x and y-axis of the input data to (256, 256). For stage 2 (tumor and cyst segmentation), the respective regions of the preprocessed image (512, 512) is cropped according to the predicted labels (masks) from stage 1.

## 2.3 Proposed Method

### 2.3.1 Architecture

Our proposed network uses the Residual 3D U-Net [5,6,7], which has four up-sampling layers and four down-sampling layers. Each layer is composed of 3D convolution, ReLU activations, and batch normalization. The first

level of the UNet extracts 32 feature maps in the proposed pipeline, and each down-sampling process maximizes the extracted feature maps up to 512. The network learning rate is 0.001; the batch size is 8, epochs are 200, and cross-entropy is used as the loss function. We utilize the Adam optimizer as the network optimizer to train the network. The proposed architecture is used for each stage separately. The Pytorch1.9.0 framework is used to implement our proposed approach.

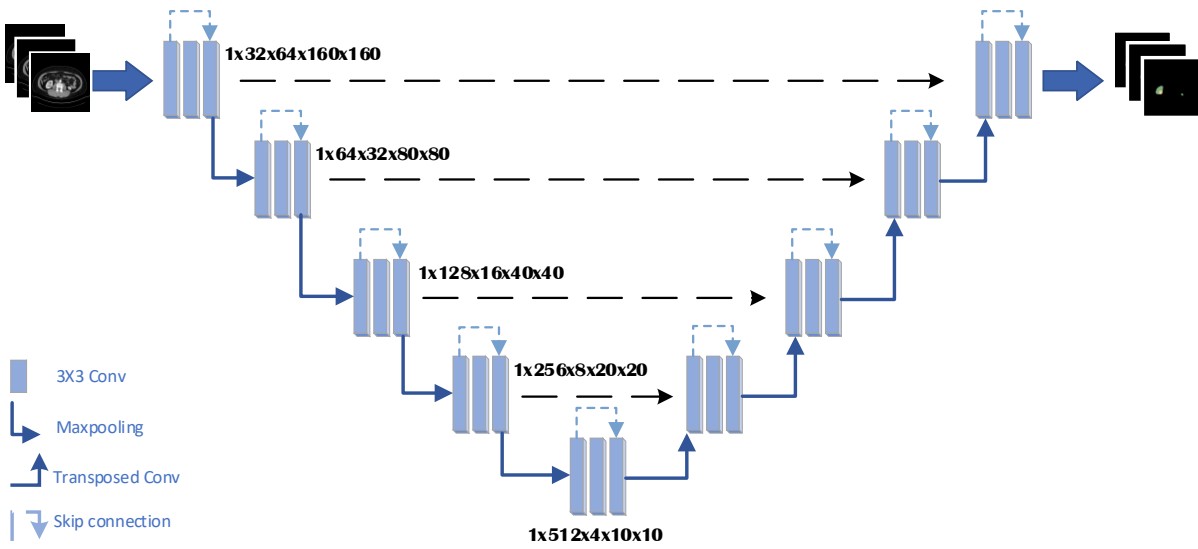

Fig. 2. Our proposed ResU-Net3D network architecture

### 2.3.2 Methodology

Stage 1: Kidney Segmentation

In the kidney segmentation stage, we initially aim to extract the whole kidney. Because the dataset contains multiple classes such as kidney, tumor, and cyst, thus it is not easy to directly detect or segment the kidney. Therefore, we change the individual kidney, tumor, and cyst masks into a single class, i.e., kidney. Considering the z-axis, we crop the 3D cube with length 96 and stride 48 by using the center crop on the x and y-axis to clip the subsequent cube with shape (1,96,192,192) and provide that as input to the kidney segmentation network. For this purpose, we use the regionprops function to analyze all input data's kidney regions and try different shape sizes, where we find the shape size (1,96,192,192) is a perfect range to cover both kidney regions entirely.

Next, we embed our proposed network architecture in the kidney segmentation stage. The kidney segmentation network predicts the renal contour. Prior to the tumor segmentation, the predicted renal is cropped and enhanced. The cropping step is essential to filter out all unnecessary information, such as the outer side of the kidney contours. The cropping is achieved by multiplying the input image with the prediction mask of predicted renal (output of stage 1).

Stage 2: Tumor and Cyst Segmentation

The tumor and cyst segmentation network take the two-channels input image by concatenating the predicted/segmented renal from stage 1 and enhanced renal after stage 1. On the z-axis, we crop the 3D cube with length 64 and stride 16 by using the center crop on the x and y-axis to clip the next cube, which comprises a single kidney with shape size (2,64,160,160) and provide that as input to the tumor and cyst segmentation network. Note that we utilize the same regionprops function to choose our clipping range. Finally, the designed network predicts tumor and cyst.

## 3 Results

We validated our approach using 30 KiTS Challenge CT images. The quantitative results are reported in Table 1. Our proposed method achieved kidney dice of 0.96. Moreover, we evaluated our model with and without HE enhancement. Applying the HE processing step results in better tumor dice and cyst dice, as shown in Table 1. Figure 3 provides the visual predictions of our proposed two-stage segmentation model.

**Table 1**. The experimental outcomes on validation data with and without HE processing.

|  | kidney | Tumor | Cyst |
|---|---|---|---|
| **Dice with HE** | 0.96 | 0.8150 | 0.4504 |
| **Dice without HE** |  | 0.6710 | 0.4028 |

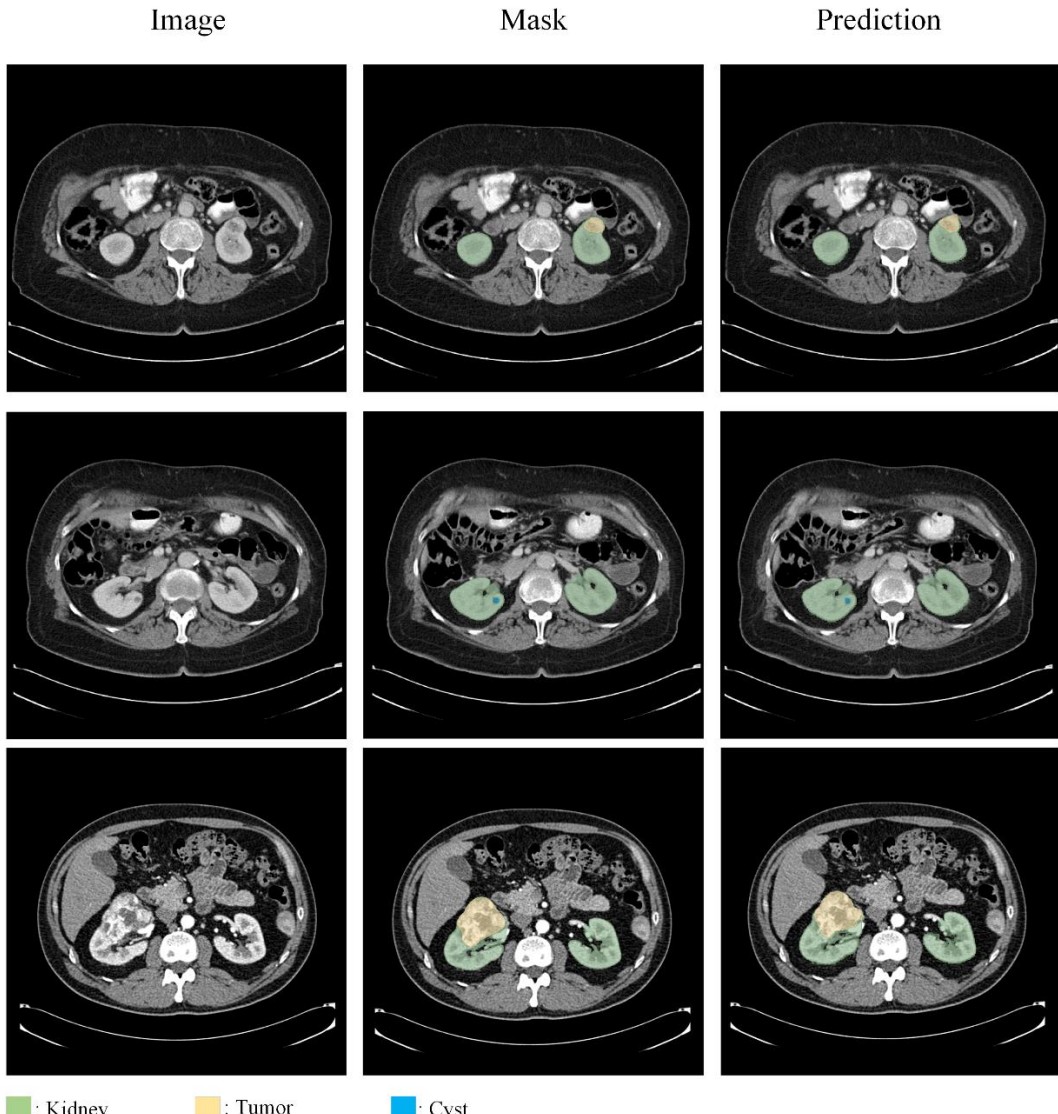

Image        Mask        Prediction

■ : Kidney     ■ : Tumor     ■ : Cyst

Fig. 3. Visual predictions of our proposed model, where the first column represents input images, column two is the ground truth (mask) images, and the third column shows the predictions of our cascaded 3D model. The colored boxes indicate each class, such as kidney, tumor, and cyst.

# 4 Discussion and Conclusion

In this manuscript, we proposed a two-stage cascaded approach to segment kidney, renal tumor, and renal cyst. For this purpose, we employed a Residual 3DUnet architecture embedded into each stage of the cascaded pipeline. Our proposed model achieved promising segmentation results in terms of kidney and tumor segmentation. Since the boundary between the tumor and the kidney is unclear, which makes the kidney and tumor segmentation difficult. Therefore, we used histogram equalization to enhance the output from the initial stage, which serves as a second channel and enriches the image information for subsequent stage 2.

Moreover, adopting the cascading strategy and training the models separately makes distinguishing between tumor and cyst easier. Our model prediction for cyst is lower as compared to the two other segmentation tasks. As many cases have no or unclear cysts. As a result, the proposed model tends to false positives predictions. To address this in the future, we will focus on model optimization and designing complex architectures that efficiently detect cysts.

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

**Comments to Editor and Reviewers**

The KiTS21 organizing committee and reviewers deserve our gratitude for dedicating their time and efforts to our paper's peer review process. We revised the whole manuscript and modified that according to the reviewer's suggestions. However, we changed our methodology by limiting our cascaded 3D model to two rather than three stages. Each step is defined clearly, and we also provide the necessary figures. Therefore, we request the organizing committee and reviewers to re-evaluate our revised paper (with a revised methodology). We will be happy to receive further suggestions and critics, if any.

## Review 1:

# Overall

- You should include references to support your statements and refer the reader to the background on the methods that you used.

  Response: We made changes to our manuscript in accordance with the provided recommendations.

# Introduction

- Please just just one new-line between these two paragraphs

  Response: We revised our manuscript and made the suggested modifications.

# Methods

- Please expand on this histogram equalization approach and provide a reference if necessary

  Response: Thank you for your thoughtful suggestions; we revised our manuscript accordingly.

- What does "HOG" stand for in the figure? Please define this acronym in the caption.

  Response: We are extremely sorry for our misdrafting. We removed the "HOG" term and replaced that with the Histogram Equalization technique.

- How did you choose your range that you clipped the images to?

  Response: We use the regionprops function to analyze all input data's kidney regions and try different shape sizes, where we find the shape size (1,96,192,192) is a perfect range to cover the both kidneys regions entirely. We also clearly mentioned it in our revised draft.

- Section 2.3.2 needs a title
  Response: We apologize for our misdrafting. We inserted a new title, i.e., 2.3.2 Methodology in the revised draft.

# Results

- The caption for table 1 is missing a space between "in" and "validation"

  Response: We revised our draft and made the suggested changes.

- Please make sure to include the final challenge results once they are known

# Discussion and Conclusion

- This section should be numbered 4, not 3

  Response: Thank you for the correction. We revised our draft and made the suggested changes.

**Rating:** 5: Marginally below acceptance threshold

**Review 2:**

The authors present a coarse-to-fine approach using different networks for each segmentation objective (i.e. each of the three classes). The paper is somewhat short on details and could definitely benefit from a figure which compares a predicted segmentation to the ground truth. Also, the authors don't mention which set of aggregated labels they trained on or what custom aggregation method they used, if any. Most used majority voting, is that what you used? Please make sure to include this in the paper.

Response: Thank you for the valuable suggestions. In the revised draft, we provided the relevant details with the figure. In section 2.1 (Training and Validation Data), we also stated that we employ voxel-wise majority voting (MAJ) for training and validation.

**Rating:** 6: Marginally above acceptance threshold