# OpenReview forum: "A Cascaded 3D Segmentation Model for Renal Enhanced CT Images"
_MICCAI.org/2021/Challenge/KiTS — Submitted to KiTS21 Challenge_

### Official Review · Reviewer_pYHL · 2021-08-30

**Rating:** 6

**Review:**

The authors present a coarse-to-fine approach using different networks for each segmentation objective (i.e. each of the three classes). The paper is somewhat short on details and could definitely benefit from a figure which compares a predicted segmentation to the ground truth. Also, the authors don't mention which set of aggregated labels they trained on or what custom aggregation method they used, if any. Most used majority voting, is that what you used? Please make sure to include this in the paper.

---

### Official Review · Reviewer_iw2J · 2021-08-30

**Rating:** 5

**Review:**

### Overall

- You should include references to support your statements and refer the reader to the background on the methods that you used

### Introduction

- Please just just one new-line between these two paragraphs

### Methods

- Please expand on this histogram equalization approach and provide a reference if necessary
- What does "HOG" stand for in the figure? Please define this acronym in the caption
- How did you choose your range that you clipped the images to?
- Section 2.3.2 needs a title

### Results

- The caption for table 1 is missing a space between "in" and "validation"
- Please make sure to include the final challenge results once they are known

### Discussion and Conclusion

- This section should be numbered 4, not 3

---

### Decision · Program_Chairs · 2021-08-30

**Decision:**

Major Revisions

**Comment:**

Please address the reviewer comments and resubmit